# Modulating the Graphitic Domains and Pore Structure of Corncob-Derived Hard Carbons by Pyrolysis to Improve Sodium Storage

**DOI:** 10.3390/molecules28083595

**Published:** 2023-04-20

**Authors:** Ning-Jing Song, Nannan Guo, Canliang Ma, Yun Zhao, Wanxi Li, Boqiong Li

**Affiliations:** 1Department of Materials Science and Engineering, Jinzhong University, Jinzhong 030619, China; 2Key Laboratory of Materials for Energy Conversion and Storage of Shanxi Province, Institute of Molecular Science, Shanxi University, Taiyuan 030006, China

**Keywords:** hard carbon, corncob core, pyrolysis temperature, microstructure, sodium-ion battery

## Abstract

Biomass-derived hard carbon materials are considered as the most promising anode materials for sodium-ion batteries (SIBs) due to their abundant sources, environmental friendliness, and excellent electrochemical performance. Although much research exists on the effect of pyrolysis temperature on the microstructure of hard carbon materials, there are few reports that focus on the development of pore structure during the pyrolysis process. In this study, corncob is used as the raw material to synthesize hard carbon at a pyrolysis temperature of 1000~1600 °C, and their interrelationationship between pyrolysis temperature, microstructure and sodium storage properties are systematically studied. With the pyrolysis temperature increasing from 1000 °C to 1400 °C, the number of graphite microcrystal layers increases, the long-range order degree rises, and the pore structure shows a larger size and wide distribution. The specific capacity, the initial coulomb efficiency, and the rate performance of hard carbon materials improve simultaneously. However, as the pyrolysis temperature rises further to 1600 °C, the graphite-like layer begins to curl, and the number of graphite microcrystal layers reduces. In return, the electrochemical performance of the hard carbon material decreases. This model of pyrolysis temperatures–microstructure–sodium storage properties will provide a theoretical basis for the research and application of biomass hard carbon materials in SIBs.

## 1. Introduction

Sodium-ion batteries (SIBs) hold great promise for a broad range of energy storage applications in the future, mainly because of their similar physicochemical properties to lithium and the abundance and low cost of sodium [1]. The anode materials in SIBs still have considerable developmental potential to enhance reversible capacity, initial Coulombic efficiency (ICE), and cycling performance compared with the more mature cathode materials [2]. As the most promising anode material, hard carbon materials have the advantage of large interlayer spaces and pores and a turbulent layer structure [3]. Furthermore, biomass-derived hard carbon materials have received significant attention due to their low cost, sustainability, and environmental benignity [4,5]. Compared with other biomass-derived hard carbons, biomass-derived hard carbon materials have been widely investigated and studied due to their wide range of sources and the high output of these materials [6].

Hard carbon is a non-graphitic carbon and presents a disordered structure in which small aromatic fragments are stacked like a house of cards in a somewhat random fashion. This random stacking creates graphitic nanodomains with several parallel or nearly parallel graphene sheets, and, at the same time, pores form in the gaps between these stacking graphene sheets [7]. The special structure causes hard carbon to have more types of sodium storage sites, which could be used for the adjustment of sodium-ion reversible capacity. The hard carbon shows a charge–discharge curve different from the other carbon-based materials, which is mainly divided into a low potential plateau region of 0.01~0.10 V and a high potential ramp zone above 0.1 V [6]. According to the “insertion-pore filling’’ mechanism, the slope region is designated as sodium ions inserted in the graphite layer, and the plateau region is filled with sodium ions and electroplated into the pores [7]. Thus, it is crucial to regulate the graphitic domains and pore structures of hard carbon materials in order to achieve high sodium storage properties [8,9].

The graphite-like domains and pore structure of hard carbon play pivotal roles for sodium storage, in which graphite-like domains act as water pipes for sodium transportation, and the pores act as a reservoir for sodium storage [10]. Therefore, optimizing the graphite-like nanocrystal structure, the porosity and pore-size distribution of hard carbon are the feasible strategies to obtain high sodium storage performance. Until now, the main method for preparing plant-derived hard carbon materials has been pyrolysis. In general, the graphitic nanodomains could be regulated by changing the pyrolysis temperature [11]. However, not all graphite-like nanocrystals are favorable to the sodium storage properties of hard carbon materials. The graphite-like nanocrystals with d-spacing of 0.36~0.40 nm are accessible for Na^+^ insertion/extraction [12]. The graphite-like nanocrystals with a d-spacing lower than 0.36 nm are too small to be accessible for Na^+^ storage [13]. Therefore, the graphite-like nanocrystals with appropriate d-spacing and content are very important to the slope capacity of hard carbon materials. It is well known that the pore structure of materials plays an important role in the diffusion and storage performance of metal ions in materials, so the design and preparation of porous materials have attracted much attention. The template method and the molten salt method are common methods for preparing porous materials [14,15,16,17]. The traditional activation method could generate new pores and further expand existing ones, ultimately increasing the specific surface area of the material [18]. However, these methods all require the introduction of additional substances during heat treatment to achieve pore structure regulation. In fact, careful study reveals that the carbon atoms could rearrange themselves with the heteroatom removal during the pyrolysis process, and this could cause the graphite-like microcrystalline size to rise, the order degree to increase, and the porosity and pore size distribution to change [13,19]. What is more, not all sizes of pores are beneficial to the sodium storage. Le et al. indicated the micropores hinder ion diffusion and hardly ever accommodate Na ions, while mesopores facilitate Na ions’ intercalation [20,21]. Therefore, the pyrolysis temperature becomes the critical factor in regulating the turbostratic nanodomains and proper pores structure, which further affects the Na-ion storage performance. However, there is scant literature on the relationship model between heat treatment temperature, microstructure, especially turbostratic nanodomains and pore structure, and the sodium storage properties of hard carbon materials.

As a high-yield agricultural byproduct, corncob is processed into hard carbon as anode material of SIBs, which not only reduces environmental pollution, but also realizes waste recycling. So far, Pin Liu et al. have prepared the corncob-based anode material of SIBs, and proved that pyrolysis temperature has a great influence on its electrochemical performance [4]. Our team has demonstrated that using graphite catalysis at the same temperature can further improve order degree and enhance the sodium storage performance of corncob-based hard carbon [22]. However, the evolution of pore structure and turbostratic nanodomains during pyrolysis, their mutual influence mechanism, and their effect on the electrochemical properties of biomass-derived hard carbon materials have not been systematically discussed.

Above all, this paper will start from the influence of the pyrolysis process in the range of 1000~1600 °C on the pore structure and microcrystalline state of corncob-derived hard carbon material, and will further explore their influence mechanism on the sodium-storage properties. In this system, the hard carbon pyrolysized at 1400 °C receives the proper structure parameters such as the d-spacing of graphite microcrystal layers, the long-range order degree, and a wider and larger pore size distribution, which exhibits the best sodium storage performance including reversible capacity, ICE, rate capability, and cycling performance. This study reveals the regulation rule of pyrolysis temperature on material microstructure and the structure–activity relationship between material structure and electrochemical properties, which could provide a reference for the subsequent research of biomass materials.

## 2. Results and Discussion

As a waste biomass byproduct, corncob could be pyrolyzed to obtain corncob-derived hard carbon for use as the anode material of SIBs, which reduces air pollution and realizes recycling. As shown in Figure 1a, the corncob-derived hard carbon materials were prepared using washing, drying, mashing, and pyrolysis processes. After pyrolysis, the color of the corncob turns from white to black, indicating a successful transition from corncob to hard carbon material, which exhibit thin sheet-like carbon architectures with oval holes. With pyrolysis at different temperatures, the size and number of holes in the carbon sheet is different. In a comparison, the number of oval macropores in CC-1400 (Appendix A) with a size of 1.0~1.5 μm in width and 2.1~3.2 μm in length was larger than in the other samples, which may be attributed to the dissolution of impure elements such as potassium in the raw corncob powder.

In order to study the effect of pyrolysis temperature on the microstructure of hard carbon materials, high resolution TEM was performed. As shown in Figure 1b–d, the CCs present turbulent structures, exhibiting a feature of hard carbon materials described in a card model [7], which included disordered phases and graphite-like phases. With the increase in pyrolysis temperature from 1000 to 1400 °C, the graphite-like microcrystallites constructed by several short-range parallel carbon hexagonal layers increase evidently, and long-range ordered structures have been formed. Meanwhile, some pores may coalesce to form larger-size pores. As the pyrolysis temperature rises further to 1600 °C, the carbon layer becomes more curved, and the interlayer distance increases. Meanwhile, some smaller-size mesopores were formed [23]. Thus, CC-1600 shows a pore size distribution similar to that of CC-1000 and CC-1200.

XPS was performed to characterize specimens to further explore the influence of pyrolysis temperature on the element and functionalities on the surface of the as-obtained hard carbon materials. As shown in Figure 2a, the peaks of the XPS wide spectrum located at 284.4 and 532.5 eV correspond to C1s and O1s, respectively. The C atom percent on the surface of samples calculated from XPS is shown in Figure 2b. The C atom percent increases from 92.37% to 95.45% when the pyrolysis temperature increases from 1000 °C to 1400 °C; correspondingly, the O atom percent decreases from 7.20% to 2.87% (Appendix A). Meanwhile, all the hard carbon materials contain negligible amounts of N, which may be derived from corncob raw materials. The C1s peak was fitted as four peaks: C=C (284.6 eV), C-C (285.1 eV), C-O (286.5 eV), and C=O (290.0 eV). As shown in Figure 2c–f, the content of C=C increases from 25.80% for CC-1000 to 29.91% for CC-1400, while the content of C-C decreases from 44.56% for CC-1000 to 42.32% for CC-1400, and the content of C=O decreases from 8.96% for CC-1000 to 7.24% for CC-1400 (Appendix A), which indicates that the increase of C=C may be due to the fact the components of sp3 hybrid carbon (usually located at defect sites of hard carbon [13]) and some carbon–oxygen polar bonds decrease with increasing pyrolysis temperature. However, as the pyrolysis temperature further rises to 1600 °C, the C atom percent reduces to 94.89% accompanied by an increase in the O atom percent to 5.02%; the content of C=C lowers to 26.65%; C-C improves to 48.30%; and C=O further decreases to 4.37% (Appendix A), which may be due to a violent rearrangement of carbon atoms that causes some oxygen-containing functional groups to reform on the surface of the material.

Raman and XRD were performed to further explore the influence of pyrolysis temperature on the microcrystalline state of the as-obtained hard carbon materials. As shown in Figure 3a, the Raman spectra of the samples exhibit broad D-bands at approximately 1340 cm^−1^ and G-bands at approximately 1580 cm^−1^, which could be attributed to amorphous carbon and crystalline graphite, respectively. The D band is associated with the vibrations of carbon atoms with dangling bonds for the in-plane terminations of disordered graphite. The G band is characteristic of the vibration of sp^2^-bonded carbon atoms in a 2D hexagonal lattice. As shown in Table 1, the relative intensity of the D and G bands (*I*_D_/*I*_G_) decreases from 2.11 for CC-1000 to 1.75 for CC-1600 with increasing pyrolysis temperature, further confirming a tendency towards ordered carbon structure.

The XRD patterns of the samples are displayed in Figure 3b. The broad diffraction peaks at 2θ = 20~25° assign to the (002) Bragg diffraction of graphitic domains, indicating all samples have been converted to hard carbon. All the interlayer distances (d_002_) of CC samples (3.88~4.41 Å) calculated from the 2θ degree of the (002) peak are larger than the size of the sodium ion (1.96 Å) (Table 1). The larger interlayer distance of carbon material is favorable to Na^+^ insertion/extraction and improves the sodium storage capability. With the increase in pyrolysis temperature (1000~1400 °C), the interlayer distance of hard carbon materials decreases from 4.03 Å to 3.88 Å, which could be attributed to the removal of heteroatoms and the rearrangement of carbon atoms, forming a more ordered graphite-like lamellar structure. This could be confirmed by XPS and Raman results. However, when the pyrolysis temperature rises to 1600 °C, the interlayer distance reversibly increases to 4.41 Å, which may be attributed to the reattachment of oxygen atoms. This is in line with the XPS result.

The shape of (002) peaks of hard carbon obtained at different pyrolysis temperatures is asymmetrical, which may be due to the mixing of graphite-like microcrystals and disordered regions [24]. To further identify the evolution of the microcrystalline phase with pyrolysis temperature, the (002) peaks are simulated by a profile-fitting process (Figure 3c–f) [13]. The corresponding diffraction angle, layer spacing, and phase ratio from main peak and fitting peaks are shown in Table 1. Combined with HRTEM analysis results, more detailed structural information on hard carbon materials can be estimated through reasonable analysis of the above fitting maps. With the pyrolysis temperature increasing from 1000 to 1400 °C, the diffraction of the disordered phase increases slightly, the interlayer distance decreases from 4.30 to 4.11 Å, and the disordered phase ratio is reduced from 47.40 to 44.76%. Meanwhile, the graphite-like phases for the samples predominate, the interlayer distance of graphite-like phases decreases from 3.65 to 3.60 Å, and the fraction of the graphitic phase increases from 52.60 to 55.24%. As the pyrolysis temperature further increases to 1600 °C, although the layer spacing reverses, the proportion of disordered phase further decreases, and the proportion of the graphite-like phase continues to increase, which is consistent with the results of XPS and Raman.

In summary, with the increase in pyrolysis temperature, the disordered phase gradually transforms into the graphite-like phase, and the d_002_ of the graphite-like phase and disordered phase first decreases and then increases, reaching the minimum value when the pyrolysis temperature is 1400 °C. For these as-prepared samples, the majority is graphite-like phase. Combined with XPS and Raman results, when the pyrolysis temperature is lower than 1400 °C, oxygen-containing functional groups are removed with the increase in pyrolysis temperature, leading to the reduction in layer spacing; meanwhile, small graphite-like phase regions merge, and more disordered phases transform into graphite-like phases. However, when the pyrolysis temperature is higher than 1400 °C, the carbon atoms undergo severe rearrangement, and the graphite-like layer is bent, resulting in increased layer spacing and a graphite-like phase.

In the Raman spectrum, *I*_D_/*I*_G_ of CC-1600 is smaller than that of CC-1400. It indicates that, with the pyrolysis temperature further rising to 1600 °C, the ordering degree of the material further improves, which may be caused by the reduction in carbon atoms with dangling bonds, or the increase in sp2 carbon atoms in the hexagonal lattice. Combined with the XRD (002) peak fitting results and TEM micrographs, the graphite-like region of CC-1600 grows with a more obvious long-range ordered structure, which can further confirm the results of the Raman spectroscopy. For CC-1600, however, HRTEM images show that more graphite layers in the graphitized region become more long-range ordered, but at the same time they become more curved, which forms more holes and presents a higher specific surface area in the process. What is more, the fitting results of XPS carbon peaks are shown in Appendix A. The dangling bond mainly contains C-O and C=O, and the sp2 carbon in the hexagonal lattice mainly refers to C=C. Accordingly, the ratio of (C-O+C=O) to C=C is 0.85, 0.83, 0.78, and 0.75 for CC-1000, CC-1200, CC-1400 and CC-1600, respectively. This trend is in line with *I*_D_/*I*_G_.

During the whole pyrolysis process, change in the microcrystal structure is accompanied by change in the ordered carbon layer, including the length, thickness, and layer spacing of the ordered carbon layer, which ultimately affects the pore structure. In order to verify the pore structure changes exhibited in HRTEM micrographs more directly, nitrogen adsorption/desorption tests were performed and are displayed in Figure 4. As shown in Figure 4a, nitrogen adsorption/desorption isotherms of CC-1000 correspond to I type curves (IUPAC classification) with a sharp uptake at an ultralow relative pressure (P/Po < 0.01), which indicates that the samples mainly comprise micropores. Meanwhile, CC-1200, CC-1400, and CC-1600 exhibit a type Ⅳ isotherm with a hysteresis loop at P/Po of 0.5~0.9, which indicates the existence of mesopore [14]. Correspondingly, BET specific surface areas (SSA) of CC-1000, CC-1200, CC-1400, and CC-1600 are 393.84, 7.13, 6.05, and 17.72 m^2^/g, respectively. Obviously, with the increase in pyrolysis temperature, the SSA of the sample decreases first and then increases. CC-1400 receives the minimum SSA value, which is very favorable to obtaining a high ICE by reducing the consumption of Na^+^ in SEI film formation [25]. The pore size distribution curves based on the NLDFT model are given in Figure 4b, which indicates the variation of micropores and mesopores with the change in pyrolysis temperature. CC-1000 possesses a large number of micropores, while CC-1200, CC-1400, and CC-1600 exhibit a large number of mesopores. The mesopores’ volume, with pore size ranging from 2 nm to 30 nm, is in the following order: CC-1600 > CC-1200 > CC-1400 > CC-1000.The pore size of CC-1200 and CC-1600 is mainly concentrates on the range of 2~10 nm, while the pore size of CC-1400 is larger than 9 nm.

The curves of the cumulative pore volume and surface area vs. the pore size are displayed in Figure 4c and Figure 4d, respectively. For CC-1000, a sharp increase in cumulative surface area and pore volume occurs at pore sizes below 2 nm, but no obvious growth at above 2 nm, indicating nearly all the pores concentrate in the range of 0.8~2 nm. In contrast, for CC-1200, CC-1400, and CC-1600, a negligible accumulation of surface area and pore volume is observed below 2 nm, and a gradual rise in pore volume at the range of 2~30 nm (4.6~20 nm for CC-1200, 9~20 nm for CC-1400, and 2.5~20 nm for CC-1600) indicates that as-obtained samples pyrolyzed above 1200 °C contain abundant mesopores with wide pore size distribution. CC-1400 exhibits a smaller mesopore volume (0.017 cm^3^/g) than CC-1200 (0.021 cm^3^/g) and CC-1600 (0.049 cm^3^/g). In summary, CC-1400 forms a pore structure with larger pore size (larger than 9 nm), smaller mesoporous volume, and wide pore size distribution, which may be conducive to sodium insertion/extraction.

During the pyrolysis process, the evolution of graphite-like layers and pore structure occurs simultaneously. The structure of the graphite-like layers not only determines the slope capacity of the hard carbon material, but also affects its platform capacity because sodium ions need to diffuse between the graphite layers and fill the pores. Compared with traditional pore-making methods [14,15,16,17,18], the pore structure is regulated only by pyrolysis. On the one hand, in addition to eliminating the need for additional pore-forming agents, controlling the pyrolysis temperature can also lead to smaller and wider pore sizes. On the other hand, the evolution of pore structure and graphite-like layers forms a synergistic effect on the electrochemical performance of hard carbon materials, which may lead to a simultaneous increase or decrease in slope capacity and plateau capacity.

Figure 5 displays the cyclic voltammograms (CV) curves of CCs in the initial four cycles at a scan rate of 0.25 mV/s within a voltage range of 0.01~3.0 V. As indicated in Figure 5a–d, all the electrodes exhibit a sharp cathodic peaks at ~0.1 V, and a broad peak between 0.5~1.2 V appears in the first discharge step; however, they are weakened and overlapped in the subsequent cycles. This phenomenon is generally explained by the irreversible reactions mainly due to the formation of solid-electrolyte interphase (SEI) film [11]. The peak at 0.1 V generally corresponds to the filling of sodium in pores, and the wide peak of 0.5~1.2 V is related to the storage of sodium between graphite layers [2,4]. For the anodic process, the peak centered at 0.1 V indicates the Na^+^ extraction from the carbon electrode. Compared to other materials, CC-1400 shows more pronounced peaks near 0.8 V and 0.01 V, indicating excellent plateau and slope capacity. In addition, the initial irreversible capacity of CC-1400 is significantly lower than that of other hard carbons, indicating that CC-1400 has a higher ICE.

EIS spectra of the obtained hard carbon materials over the frequency range from 100 kHz to 0.01 Hz were shown in Figure 6. The high-frequency semicircle is associated with the resistance of the SEI layer (R_SEI_), the middle-frequency semicircle is ascribed to the charge transfer resistance through the electrode/electrolyte interface (Rct), and the slope of the straight line in the low-frequency region refers to the diffusion condition of Na^+^ ions in a solid-state compound [26,27]. As shown in Figure 6, compared with other materials, CC-1400 displays a smaller R_SEI_, suggesting it could exhibit high ICE. Meanwhile, CC-1400 exhibits a smaller R_ct_, which indicates that charge transfer is easier at the CC-1400 electrode and electrolyte interface. What is more, it presents a higher slope straight line at low frequencies for CC-1400, demonstrating that Na^+^ has a faster diffusion rate in CC-1400. In summary, CC-1400 presents a better electrochemical performance.

Figure 7a shows the Coulombic efficiency of discharge/charge profiles of CCs at a current density of 30 mA/g. This large initial irreversible capacity loss is mainly caused by the initial formation of a solid electrolyte interphase (SEI) [28]. However, with the pyrolysis temperature rising from 1000 to 1400 °C, the ICE increases from 52.56% to 66.59%, which may be attributed to the following: (1) the rich graphite nanocrystals decreasing the consumption of Na ions during the SEI formation; (2) the decreasing ‘trap effect’ of residual oxygen groups and defects on Na ions [29]; (3) the larger size of the hole facilitating sodium insertion/extraction to reduce dead sodium formation. Meanwhile, when the pyrolysis temperature increases to 1600 °C, the ICE decreases to 41.13% owing to the larger SSA and the smaller size of pores [30]. The Coulombic efficiency increases rapidly upon cycling and reaches nearly 100% after the initial few cycles, which may be due to the formation of stable SEI.

As shown in Figure 7b, the galvanostatic discharge/charge reversible capacity of the second cycle is divided into two parts. Above the inflection point (0.1 V) is considered as the slope part and below the point (0.1 V) is classified as the plateau [31]. The inset in Figure 5b provides the detailed slope/plateau capacity values: the slope/plateau capacity is 67.75/63.22 mAhg^−1^, 91.40/126.52 mAhg^−1^, 117.44/180.56 mAhg^−1^, and 42.32/29.48 mAhg^−1^ for CC-1000, CC-1200, CC-1400, and CC-1600, respectively. Obviously, the slope/plateau capacity of CC-1400 is superior to the others. The slope capacity is related to the insertion of sodium between parallel or nearly parallel graphite layers [7,31]. With the pyrolysis temperature increasing from 1000 °C to 1400 °C, the formation of increasing numbers of graphite-like layers and longer-range order graphite microcrystals favors the intercalation of sodium, and, as a result, the slope capacity increases. However, as the pyrolysis temperature rises further to 1600 °C, the slope capacity reduces owing to the reduced number of graphite layers due to the rearrangement of carbon atoms and the bending of graphite layers. The plateau capacity is contributed to the insertion of the metal into the pores between randomly stacked layers [7,32]. Compared with the slope capacity, the plateau capacity shows a more visible change. As the pyrolysis temperature increases from 1000 °C to 1400 °C, the specific surface area of hard carbon materials rapidly lowers from 393.84 to 6.05 m^2^/g, and the micropores (of sizes less than 2 nm) significantly decrease; meanwhile, the mesopore size gradually increases from 2~4 nm for CC-1000 to larger than 9 nm for CC-1400, and the pore size distribution of CC-1400 is more uniform, which favors sodium storage. Therefore, as the pyrolysis temperature increases from 1000 °C to 1400 °C, the plateau capacity improves remarkably. As the pyrolysis temperature further rises to 1600 °C, the graphite layers begin to bend, the specific surface area is enhanced, and the pore size decreases to the range of 3~11 nm. As a result, the sodium storage performance in the plateau region of CC-1600 reduces.

In order to further explain the influence of the evolution of the overall microstructure of hard carbon materials on the properties of sodium storage, this phenomenon was further analyzed later. In Figure 4, CC-1600 presents the largest pore volume, with pore sizes concentrated in the range of 2~10 nm. If the graphite-like layer structure of all the obtained hard carbon material is same, the larger size and volume of the pore may be beneficial to the sodium storage performance of the hard carbon material, but it is not favorable to the ICE of the hard carbon material. However, with the change in pyrolysis temperature, the evolution of graphite-like layer structure is integrated with the change in pore structure. In the process of sodium-ion storage, sodium ions need to be inserted into the graphite-like layer first before reaching the pores. The former mainly affects the slope capacity of the material, while the latter mainly affects the plateau capacity of the material. Compared with CC-1400, there are more graphite-like phases and fewer disordered phases in CC-1600. At the same time, d_002_ and oxygen content increases, indicating that the degree of order for graphite-like phases in CC-1600 reduces, which is consistent with the TEM results. Due to the rearrangement of carbon atoms, the graphite-like layers of CC-1600 are further bent, which seems to promote the formation of a wider pore size distribution and is conducive to the platform structure of CC-1600. However, the reduced order degree of the graphite-like layer inhibits the growth of slope capacity and also restricts the entry of more sodium ions into the pores, which ultimately leads to the degradation of the sodium storage performance of the material.

As shown in Figure 7c, the cycling performance of CCs is tested at a constant current density of 30 mA/g. All corncob-derived hard carbon material electrodes exhibit excellent cycling stability. The initial discharge specific capacity of CC-1000, CC-1200, CC-1400, and CC-1600 is 128.25 mAhg^−1^, 204.59 mAhg^−1^, 272.06 mAhg^−1^, and 61.46 mAhg^−1^, respectively. After 100 cycles, the specific capacity retains 88.77%, 81.43%, 78.52%, and 94.94% of the initial specific capacities, which are 113.85 mAhg^−1^, 166.59 mAhg^−1^, 213.63 mAhg^−1^, and 58.35 mAhg^−1^, respectively. During the cycling process, CC-1400 retains the highest specific capacity, which may benefit mainly from the effective penetration of electrolyte and reduced strain during cycling via porous skeleton with wide distribution and larger size [19].

The rate capability is assessed at the current densities from 30 mA/g to 2.0 A/g to evaluate the dynamic performance of the CCs’ electrodes. As shown in Figure 7d, CC-1000 exhibits a reversible specific capacity of 125.23, 112.66, 76.70, 43.77, 33.04, 22.29, and 72.52 mAhg^−1^ at the current rate of 30 mA/g, 60 mA/g, 200 mA/g, 500 mA/g, 1.0 A/g, 2.0 A/g, and 200 mA/g, respectively. At the same current densities, CC-1200 receives a reversible specific capacity of 198.49, 137.13, 81.84, 59.33, 43.37, 24.61, and 79.71 mAhg^−1^, respectively. CC-1400 delivers a reversible specific capacity of 267.06, 237.31, 169.48, 90.59, 57.64, 35.44, and 165.07 mAhg^−1^ at the current rate of 30 mA/g, 60 mA/g, 200 mA/g, 500 mA/g, 1.0 A/g, 2.0 A/g, and 200 mA/g, respectively. Meanwhile, CC-1600 exhibits a reversible specific capacity of 67.46, 63.90, 44.11, 30.65, 20.75, 11.37, and 43.30 mAhg^−1^, respectively, at the same current densities. All CCs show moderate rate capacity, and the rate capacity of CC-1400 still achieves the maximum value, which indicates the excellent rate performance of CC-1400. It may be attributed to the more ordered graphite microcrystalline structure and the pore structure with wide distribution and larger size. Overall, CC-1400 exhibits the best sodium storage performance considering the reversible capacity, ICE, rate capability, and cycling performance.

## 3. Experimental Section

The innermost soft parts of the corncob (Weihai City, Shandong Province, China) were scooped out and collected, followed by washing in deionized water, drying in a blast oven at 45 °C for 6 h, and mashing into fine powder. The fine white powder was firstly pyrolyzed at 500 °C with a heating rate of 3 °C/min and maintained for 2 h in a furnace under Ar atmosphere. These products were marked as CC-500. Then, CC-500 was further heated up to 1000 °C, 1200 °C, 1400 °C, and 1600 °C, respectively, at a rate of 5 °C/min under Ar atmosphere and pyrolyzed at a constant temperature for 2 h. Finally, the samples were impregnated in 3 mol/L of HCl solution for 24 h and followed by washing with deionized water until their pH reached 7. The washed samples were freeze-dried for 24 h and denoted as CCs, such as CC-1000, CC-1200, CC-1400 and CC-1600, respectively.

The as-obtained samples were characterized by using a scanning electron microscope (SEM, JSM-7001F, 3.0 kV, JEOL, Japan), transmission electron microscopy (TEM, JEM-1011, JEOL, Japan), an X-ray diffractometer (XRD, Cu Ka radiation, D8 Advance, BRUKER/AXS, Germany), a Raman spectrophotometer (LabRAM HR Evolution, Horiba, France), and an X-ray photoelectron spectrometer (XPS, Thermo ESCALAB 250XI, Thermo Fisher Scientific, USA). The specific surface area was measured using a Micromeritics APSP 2460 surface area analyzer (BET, ASAP2460, Micromeritics, USA).

The electrochemical properties were measured using coin-type (CR2032) half cells. To produce a slurry, 80 wt.% samples, 10 wt.% acetylene black, and 10 wt.% sodium carboxymethyl cellulose (degree of substitution: 0.85–0.95, viscosity, 1%, 25 °C: 2500~3000 mPa·s) were mixed in deionized water. This slurry was then uniformly loaded on a Cu foil with a doctor blade to prepare a film-type electrode. The sample was dried at 80 °C under vacuum for 12 h, and then cut into circular electrodes. The loading active mass on the electrode was 0.8~1.2 mg. The cells were assembled in an Ar-filled glove box (Dellix Co., Chengdu, China) with sodium foil as both the reference and counter electrode, glass fiber as the separator, and a solution of 1.0 M NaClO_4_ in ethylene carbonate (EC): dimethyl carbonate (DMC) = 1:1 vol.% as the electrolyte. The galvanostatic charge–discharge measurements were carried out on a battery testing system (Neware Co., Shenzhen, China) in the potential range of 0.01 V to 2.5 V. The cyclic voltammograms (CVs) were tested using CHI 650E (Chenhua, Shanghai) between 3.0 and 0.01 V (vs. Na/Na^+^) at a scan rate of 0.5 mV s^−1^. Electrochemical impedance spectroscopy (EIS) was also measured on a CHI 650E (Chenhua, Shanghai) and performed in the frequency range of 10 mHz~100 kHz at an amplitude of 5 mV.

## 4. Conclusions

Corncob-derived hard carbon could be prepared by pyrolysis from corncob. Pyrolysis temperature plays a vital role for the microstructure and the electrochemical performance of hard carbon. With the pyrolysis temperature increasing from 1000 °C to 1400 °C, the number of graphite microcrystal layers rises, the long-range order degree is enhanced, more graphite-like phases form, and a pore structure with a wide distribution and larger sizes is generated. Finally, the specific capacity, the ICE, and the rate performance of hard carbon materials improve simultaneously. However, as the pyrolysis temperature further increases to 1600 °C, the graphite-like layer begins to curl, and number of graphite microcrystal layers reduces, which results in a significant reduction in the electrochemical performance of CCs. To sum up, mesopore structures with wide distribution and larger size, more graphite-like phases, fewer oxygen atoms, and a higher degree of order are beneficial to the electrochemical properties of corncob-derived hard carbon materials. This work will provide a theoretical basis for the application of biomass hard carbon materials in SIBs.

## Figures and Tables

**Figure 1 molecules-28-03595-f001:**
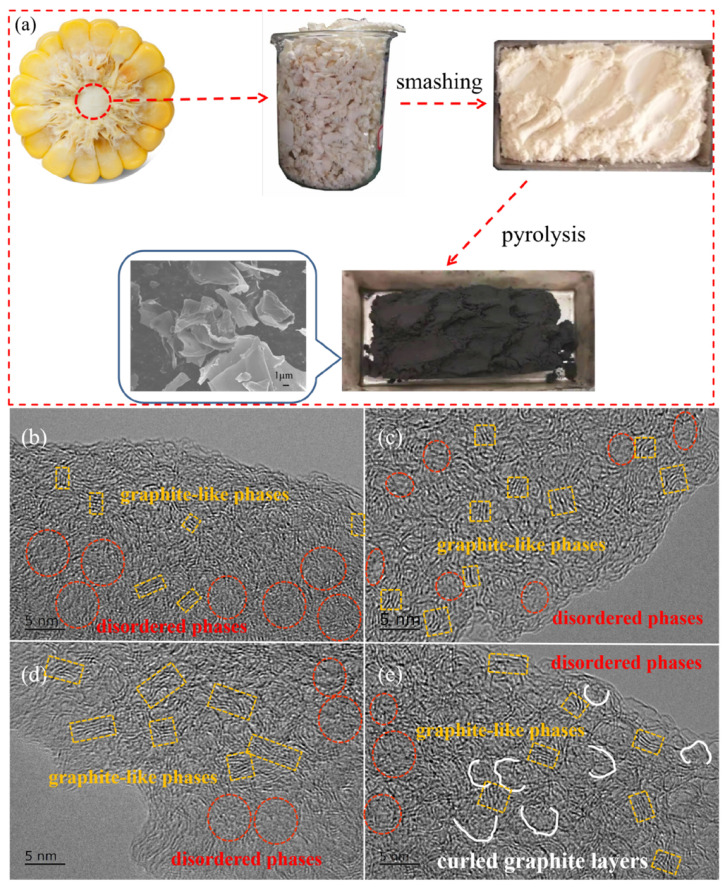
(**a**) Preparation process of corncob-derived hard carbon materials. High-resolution TEM images of CC-1000 (**b**), CC-1200 (**c**), CC-1400 (**d**), and CC-1600 (**e**).

**Figure 2 molecules-28-03595-f002:**
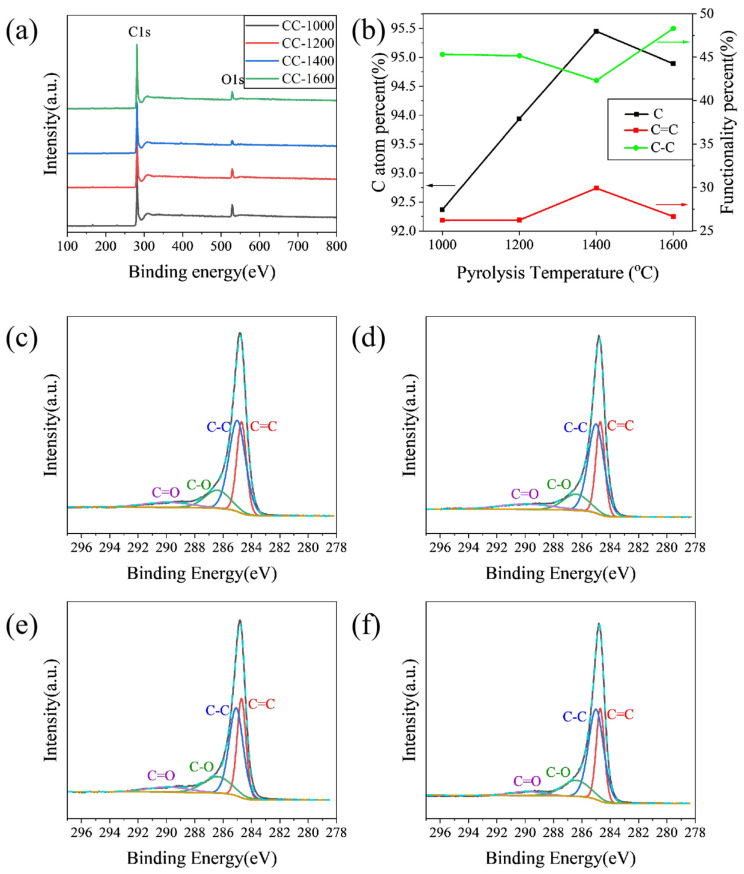
(**a**) XPS wide spectra; (**b**) functionality percent and C atom percent; high-resolution of C1s XPS spectra of (**c**) CC-1000, (**d**) CC-1200, (**e**) CC-1400, and (**f**) CC-1600.

**Figure 3 molecules-28-03595-f003:**
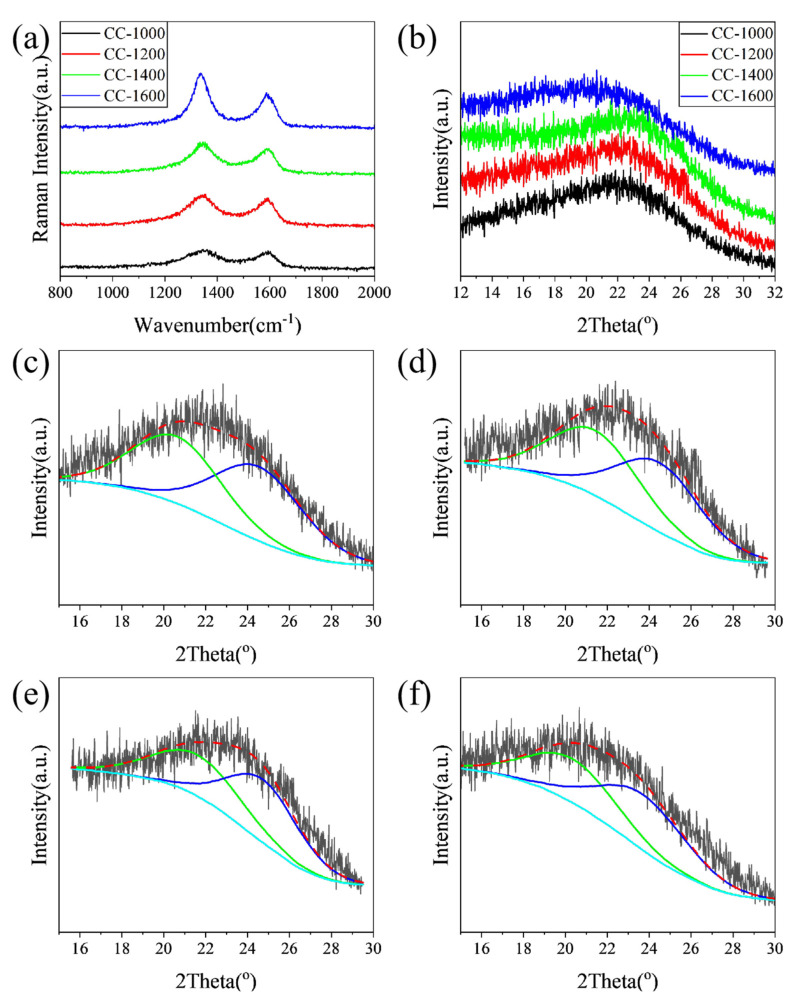
(**a**) Raman spectra; (**b**) XRD patterns of the all as-obtained samples; the (002) peak fitting curves of the XRD pattern of (**c**) CC-1000, (**d**) CC-1200, (**e**) CC-1400, and (**f**) CC-1600.

**Figure 4 molecules-28-03595-f004:**
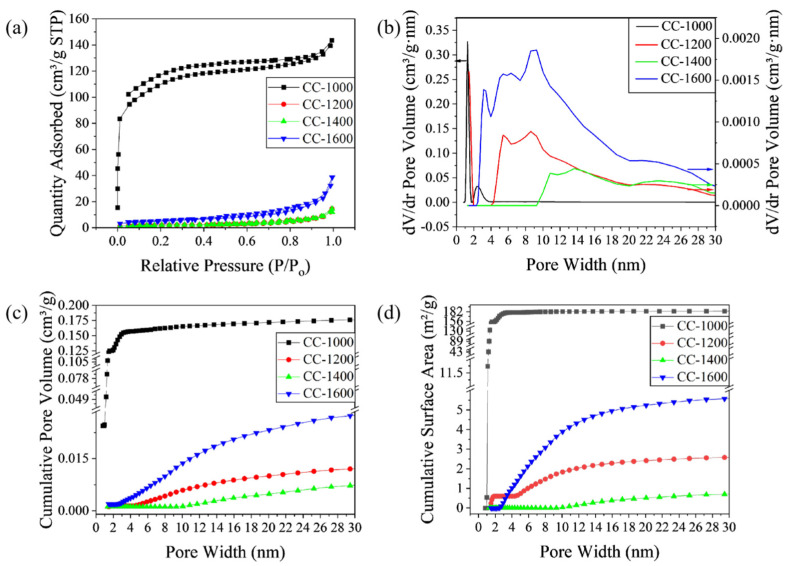
(**a**) Nitrogen adsorption/desorption isotherms; (**b**) Pore size distribution curves based on NLDFT model; (**c**) cumulative pore volume; and (**d**) cumulative surface area of different corncob hard carbon materials.

**Figure 5 molecules-28-03595-f005:**
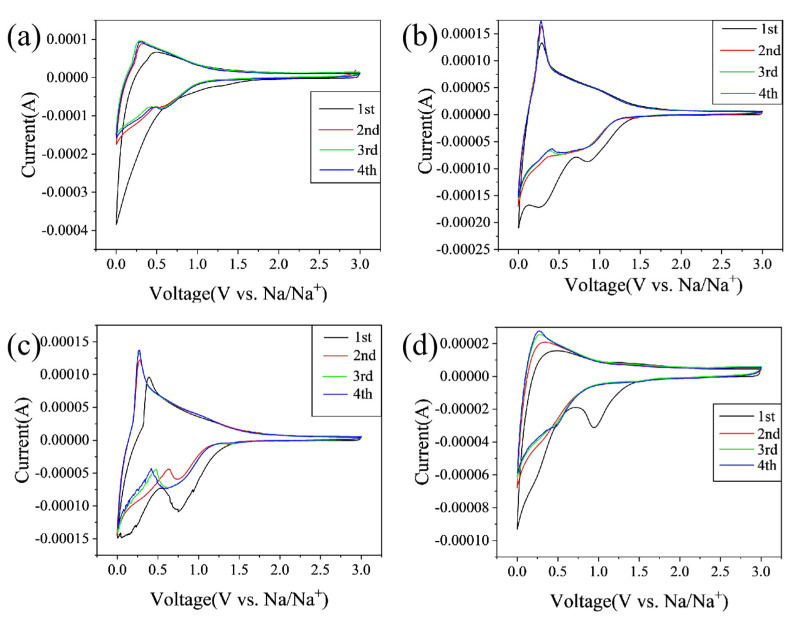
CV diagrams of (**a**) CC-1000, (**b**) CC-1200, (**c**) CC-1400, and (**d**) CC-1600.

**Figure 6 molecules-28-03595-f006:**
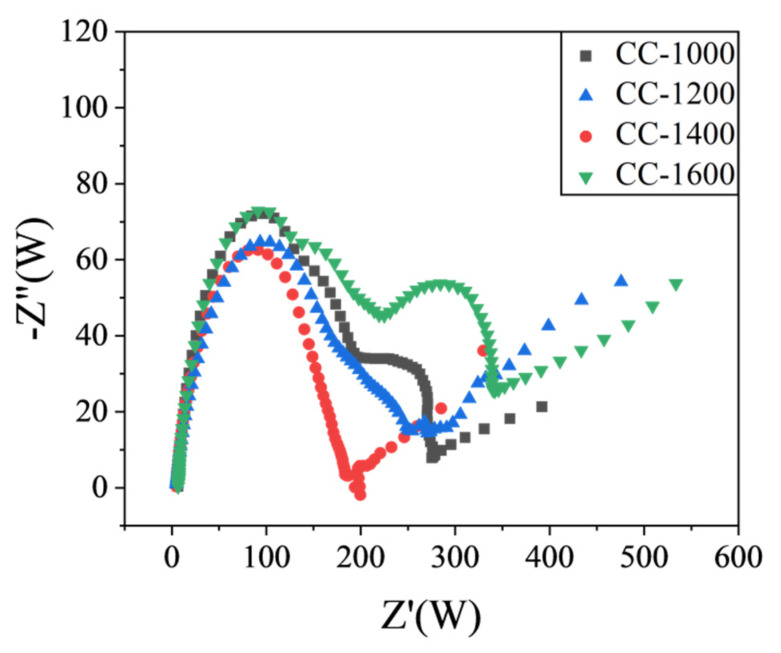
Electrochemical impedance spectra (EIS) of CCs.

**Figure 7 molecules-28-03595-f007:**
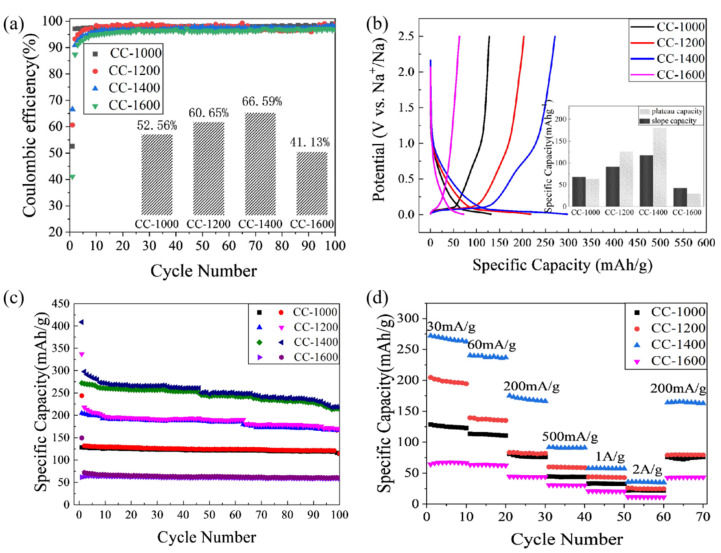
(**a**) The Coulombic efficiency of discharge/charge profiles (inset: the ICE of discharge/charge profiles); (**b**) the second galvanostatic discharge/charge profiles (inset: the plateau capacity and slope capacity of the second cycle) of CCs at a current rate of 30 mA/g; (**c**) cyclic performance; and (**d**) rate capability of CCs.

**Table 1 molecules-28-03595-t001:** Physical parameters calculated by XRD and Raman patterns of the as-obtained samples.

	*I*_D_/*I*_G_	002 Peak		Disordered Region			Graphite-like Region		
		2θ	D_002_ (Å)	2θ	d_002_ (Å)	Area (%)	2θ	d_002_ (Å)	Area (%)
CC-1000	2.11	22.04	4.03	20.62	4.30	47.40	24.34	3.65	52.60
CC-1200	1.99	22.43	3.96	21.41	4.15	46.91	24.49	3.63	53.09
CC-1400	1.77	22.90	3.88	21.61	4.11	44.76	24.74	3.60	55.24
CC-1600	1.75	22.12	4.41	20.32	4.37	44.05	23.73	3.75	55.95

## Data Availability

Not applicable.

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
