# Peer review of "Modulating the Graphitic Domains and Pore Structure of Corncob-Derived Hard Carbons by Pyrolysis to Improve Sodium Storage"

_molecules, 2023, doi:10.3390/molecules28083595_

Round 1

Reviewer 1 Report

In this work, the authors prepared hard carbon materials from biomass corncob and systematically studied the impact of pyrolysis temperature on the microstructure and electrochemical properties. The experiment findings relevant to the hard carbon anode offered in this manuscript are interesting. However, there are a few issues to be addressed before it can be accepted for publication.

1. The obtained results of CC-1600 are confusing, which shows high graphitic degree on Raman spectrum. However, its surface area is higher than that of CC-1400 and CC-1200 together with decreased content of C=C. And the O atom percent of CC-1600 is missed in the manuscript. The authors need to clarify this.

2.  The thickness of the loaded active materials is better evaluated. How about the active mass loading on the electrode?

3. The mechanism underpinning the electrochemical performance of CC-1400 has also been well clarified. To make the research more convincing, studies like EIS and/or postmortem measurements are suggested to be conducted.

4. Some research work regarding porous electrodes is suggested to be consulted, such as Sustainable Materials and Technologies 2023, 35, e00535; ACS Appl. Mater. Interfaces 2020, 12, 2, 2481–2489; J. Power Sources 2022, 514, 230593; Microporous Mesoporous Mater. 2017, 238, 78-83.

Reviewer 2 Report

This work reported a method of preparing hard carbon anode for sodium ion batteries by pyrolysis from corncob. The hard carbon anode has comprehensive properties of high capacity, high magnification and high ICE. It provides new ideas for preparing biomass-derived carbon anodes and exploring their optimal properties. This work may deserve publication in Molecules, however, some changes are required before a revised paper can be considered for publication.

Here are some important comments:

1. Similar work about hard carbon anode prepared from corncob have been reported(ChemElectroChem, 2023: e202201117. Journal of Materials Chemistry A, 2016, 4(34): 13046-13052.). What is the main difference of this work in comparison with those reported work?

2. The author claimed that some carbon-oxygen polar bonds decrease with increasing pyrolysis temperature. However, the reviewer is not convinced by the figure 2. More evidence should be provided to confirm the decrease of carbon-oxygen polar bonds.

3. The plateau capacity changes more obviously with different pyrolysis temperature. Please describe the reason for the increase of the plateau capacity with more details.

4. The author’s discussion on the rate performance which is improved with the increase of pyrolysis temperature below 1400 ℃ is inconsistent with the data shown in Figure 5.

5. Please carefully check the whole manuscript to avoid technical errors such as missed spacing.

6. Please align the legend in the figure 2a and figure 3a.

Reviewer 3 Report

This manuscript describes corncob-derived hard carbon as anode material for Sodium-ion batteries. The idea of biomass-derived hard carbon as anode materials has been previously explored by many researchers. This manuscript focuses on the effects of pyrolysis on the porous structure of hard carbon, and their relative performance in half cells. This writing does provide insights for studies on hard carbon porosity and its effects on performance. However, multiple concerns will need to be addressed/answered before it can be accepted.

Major comments:

1. Extensive English editing of the manuscript is highly recommended. There are many typos and unnatural spacings throughout the text. Please follow the standard guidelines for English writing. Some typos include but are not limited to the following: line 30, because (of); line 31, a big development opportunities; line 39, owing to (nouns); line 42, small aromatic fragment are; line 50, 'insertion-fifilling'?; line 104, amashing; line 133, obtained; line 242, based (on);

2. in experimental section, more details need to be provided, such as where the corncob was obtained, where all chemicals were obtained, and their relative purity content.

3. The authors state that SEM was used for the characterization of hard carbon, however, there is only one image of SEM in Fig. 1, with low resolution. It's recommended that authors provide more SEM images that characterize CC-500, 1000, 1200, 1400, and 1600 as necessary either in the main text or in supplementary information, and include discussions about their morphological differences.

4. The authors state that CC-1600 has curled layers, which is hard to be seen under current TEM resolution. If authors are capable of providing better res. images, please do so either in supplementary info. or main text.

5.  In Fig. 2b discussion, the authors state that C increased from 92.37 to 95.45, with O decreased from 7.20 to 2.87. Is there impurity introduced in the pyrolysis process? Or was it N? Also, I do not see the O percent graph in Fig. 2b, please consider adding it.

6. Based on Fig. 4, it seems as if CC-1600 is the best as it has the most porous structure. However according to conclusion, CC-1600 has curly platelets that hinder performance. Please include this aspect in porosity discussion.

Round 2

Reviewer 1 Report

The aurthors have addresssed the concerns. Therefore, I agree its publication in present form.

Reviewer 3 Report

After carefully reviewing the revised manuscript, I believe that the authors have addressed all my comments and comments from peer reviewers. I believe this revised manuscript is ready for publication.

Specifically, I appreciate that the authors fix the formatting and typos, and added supplementary information for SEM images and explanation for TEM images, and I also like the fact that authors added CV curves and EIS curves, which is necessary for what the authors claim.